# Tobacco Smoke Exposure in Children and Adolescents: Prevalence, Risk Factors and Co-Morbid Neuropsychiatric Conditions in a US Nationwide Study

**DOI:** 10.3390/healthcare12212102

**Published:** 2024-10-22

**Authors:** Mona Salehi, Mahdieh Saeidi, Natasha Kasulis, Tala Barias, Tejasvi Kainth, Sasidhar Gunturu

**Affiliations:** 1Department of Psychiatry, Bronx Care Health System, New York, NY 10457, USA; saleh109@umn.edu (M.S.);; 2Department of Psychiatry, Johns Hopkins University School of Medicine, Baltimore, MD 21205, USA; 3Department of Psychiatry, University of Minnesota School of Medicine, Minneapolis, MN 55455, USA; 4Research Center for Addiction and Risky Behaviors, Iran University of Medical Sciences, Tehran 14535, Iran; mahdieh.saeidi@nyspi.columbia.edu; 5Department of Psychiatry, Icahn School of Medicine at Mount Sinai, New York, NY 10029, USA

**Keywords:** tobacco smoke exposure (TSE), prevalence, co-morbidity, socio-demographics, characteristics, severity

## Abstract

Background: Tobacco smoke exposure (TSE) is a major public health concern, impacting not only smokers but also those around them, particularly children and adolescents. TSE is linked to various neuropsychiatric conditions and significantly impacts quality of life. This study examines the prevalence, socio-demographic factors, and the impact of TSE on the severity of neurological and psychiatric co-morbidities. Methods: Data from the National Survey of Children’s Health (NSCH) in the US from 2020 to 2021 were used in this study. We included 91,404 children and adolescents aged between 0 to 17 years for the TSE prevalence and socio-demographic analysis, and 79,182 children and adolescents aged between 3 and 17 years for the neuropsychiatric co-morbidities analysis. The mean age of these individuals was 8.7 (standard deviation: 5.3), and 11,751 (12.9%) had confirmed TSE. Results: Our analysis showed that TSE is more common in males (53%) than females (47%). Additionally, the odds of TSE were higher in families with a lower income level and with American Indian/Native Alaska racial descent. We found that 36.4% of youths with TSE developed at least one co-morbid condition. The most common neuropsychiatric co-morbidities were anxiety problems (15.7%), Attention-Deficit Hyperactivity Disorder (ADHD) (15.5%), behavioral and conduct problems (13.7%), and learning disability (12%). Females had lower odds of co-morbid anxiety (OR: 0.3, *p* = 0.02) and Autism Spectrum Disorder (ASD) (OR: 0.9, *p* = 0.04) than males. Asians showed lower odds of co-morbid ADHD (OR: 0.3, *p*-value: 0.001), anxiety problems (OR: 0.4, *p*-value: 0.003), speech/other language disorder (OR: 0.4, *p*-value: 0.001), developmental delay (OR: 0.4, *p*-value: 0.001), behavioral and conduct problems (OR: 0.4, *p*-value: 0.003), and learning disability (OR: 0.5, *p*-value: 0.004). Conversely, American Indian children and adolescents had higher odds of co-morbid headaches (OR: 3, *p*-value: 0.005). TSE co-occurring with Tourette’s Syndrome (TS) (OR: 4.4, *p* < 0.001), ADHD (OR: 1.3, *p* < 0.001), developmental delay (OR: 1.3, *p* < 0.001), behavioral problems (OR: 1.3, *p* < 0.001), headaches (OR: 1.3, *p* = 0.005), depression (OR: 1.2, *p* = 0.02), anxiety (OR: 1.2, *p* < 0.01), ASD (OR: 1.2, *p* < 0.001), and learning disability (OR: 1.2, *p* = 0.03) may contribute to a more severe manifestation. Conclusions: ADHD, behavioral/conduct problems, and learning disabilities were the most prevalent co-occurring conditions with TSE. Our findings show that 36.4% of youths with TSE had at least one neuropsychiatric comorbidity. Screening for these conditions in youths exposed to TSE is crucial for early detection and interventions to increase their mental health and well-being.

## 1. Introduction

Tobacco smoke exposure (TSE), a prominent public health issue, is widely recognized for its harmful effects on the health of non-smokers. According to data from the National Health and Nutrition Examination Survey, from 2013 to 2016, 35.4% of U.S. non-smoking youths aged 3–17 were exposed to second-hand smoke (SHS) [1]. Environmental TSE can come in the form of (a) second-hand smoke, which is involuntarily breathing in smoke from others who are actively smoking, and (b) third-hand smoke, involuntary smoke exposure through objects that encounter tobacco smoke contaminants such as skin, hair, furniture, clothing, and dust [2,3]. The unwanted environmental exposure to smoking, most often experienced by children, significantly impacts their mental well-being and causes an elevated risk of subsequent psychiatric manifestations [4,5]. Notably, among these children, some of the prevalent neurobehavioral issues include behavioral or conduct issues, developmental delay, intellectual disability, learning disability, and speech or other language disorders [6]. It is crucial to recognize that the impact of tobacco smoke reaches beyond smokers to those in their vicinity, often adversely affecting children [2].

In recent years, the mental health crisis among young people in the U.S. has been escalating, with 1 in 10 children experiencing severe psychiatric illness, and 1 in 5 being diagnosed with a mental health disorder each year [7,8,9,10,11]. Previous research suggests a positive correlation between TSE and the emergence of signs, symptoms, and qualifying diagnoses of neurodevelopmental and psychiatric disorders in children [12,13,14,15,16,17]. Attention deficit hyperactivity disorder (ADHD), generalized anxiety disorder (GAD), major depressive disorder (MDD), and conduct disorder (CD) are primary diagnoses that are connected to environmental TSE [18,19,20,21]. A cross-sectional study in China found a strong positive correlation between early life exposure to second-hand smoke and behaviors congruent with autism spectrum disorder (ASD) in young children, with a higher likelihood of ASD-like behaviors when tobacco exposure was more frequent and longer in duration [22].

The World Health Organization (WHO) Framework Convention (FCTC) on Tobacco Control has led to global efforts to increase public knowledge of, and education about, the negative health effects of tobacco and TSE, which has resulted in decreased numbers of household smokers overall, however, a stark disparity continues to exist between higher-income populations and those of lower-income households [23]. Second-hand smoke exposure increased with lower family income and was over three times higher in youths living with two or more smokers compared to those not living with smokers [1]. The intergenerational impact of TSE has differing adverse effects that are compounded at escalating rates in younger ages, ethnic minorities, and those of lower familial socioeconomic status [24,25]. Studies using animal models have also observed differences between the sexes in behavioral outcomes arising from TSE, such as female rats having a higher sensitivity to nicotine’s effects on locomotor activity compared to male rats [26,27,28].

This study seeks to enhance existing research on the influence of TSE on the prevalence and severity of neuropsychiatric disorders in youths. In addition, we aim to evaluate the prevalence of current neuropsychiatric conditions in a nationwide sample of youths exposed to tobacco smoke. We also explore demographic variables and sociodemographic predictors of various intersectional determinants of health, including age, gender, ethnicity, and familial socioeconomic status.

## 2. Materials and Methods

### 2.1. Study Design and Setting

This cross-sectional study utilized combined data from the 2020–2021 National Survey of Children’s Health (NSCH), an annual household-based survey conducted by the United States Census Bureau. The NSCH aims to provide comprehensive national and state-level information on the physical and emotional well-being of children aged 0 to 17 in the United States. The survey is sponsored by the Maternal and Child Health Bureau at the Health Resources and Services Administration, in collaboration with the Census Bureau, the National Center for Health Statistics at the Centers for Disease Control, the Child and Adolescent Health Measurement Initiative, and a National Technical Expert Panel.

The data collection methods involved mail- and web-based parent-proxy surveys in both English and Spanish. Funding for the survey was provided by the Health Resources and Services Administration Maternal and Child Health Bureau.

### 2.2. Sampling and Data Collection

The NSCH employed the State and Local Area Integrated Telephone Survey (SLAITS) as its sampling frame. Trained interviewers initiated random calls to identify households with at least one child under the age of 18. From these eligible households, one child was selected at random for the interview. Additionally, an interview was conducted with the adult in the household who was most knowledgeable about the selected child’s health and well-being [29].

Survey participants had the option to respond online, by mail, or over the phone. Initially, they completed a brief screener questionnaire (NSCH-S1 or S1) to determine eligibility based on the presence of children aged 0–17 in their location. Subsequently, participants filled out a detailed questionnaire about one randomly chosen child in their home. The nature of the questions varied according to the age of the selected child. Demographic details, parental health, family functioning, child health, neighborhood, and other demographics were among the topics covered in the questionnaire [30]. The list of survey questions is available in Appendix A.

The study encompassed a nationwide scope, covering all 50 states and the District of Columbia. The target population consisted of non-institutionalized children aged 0–17 years. The sample sizes for 2020 and 2021 were 42,777 and 50,892, respectively, with weighted response rates of 42.4% and 40.3% [7]. We included 91,404 children and adolescents aged between 0 to 17 years for the TSE prevalence and socio-demographic analysis, and 79,182 children and adolescents aged between 3 and 17 years for the neuropsychiatric co-morbidities analysis.

The weighting process begins with assigning base sampling weights to households, followed by nonresponse adjustments and post-stratification. Iterative raking aligns the weights with population controls, addressing demographic factors at the state, household, and child levels. Large weight trimming prevents undue influence, and population controls from the ACS ensure methodological consistency. Adjustments for demographic alignment are made to household- and child-level weights, acknowledging limitations, especially with fewer than 30 instances. The overall goal is to balance precision and reliability in weighted totals compared to population controls.

### 2.3. Ethical Considerations

Ethical approval for the NSCH study plan was granted by the Institutional Review Board (IRB) of the Centers for Disease Control and Prevention (CDC). Informed consent was obtained from parents or legal guardians of adolescents, while teenagers themselves provided their assent. It is important to note that all aspects of the research adhered to the principles outlined in the Declaration of Helsinki [31].

### 2.4. Confidentiality Measures

The NSCH prioritizes confidentiality through advanced anonymization techniques, removing direct identifiers and masking potential clues. Participants are well informed about the survey’s purpose, questions, and data use, providing voluntary informed consent. Robust data security measures, such as secure storage and encryption, minimize the risk of unintentional or malicious disclosures. Compliance with privacy laws, regular audits, and adaptability to regulatory changes ensure continuous participant confidentiality. Restricted access to identifiable information, and the provision of aggregated, de-identified data to researchers, prevent the disclosure of specific details, thereby maintaining confidentiality [32].

### 2.5. Dependent Variables

Co-morbid neuropsychiatric diagnoses and their severity were derived from the questions “Does this child currently have the condition?” and “Would you describe this child’s current condition as mild, moderate or severe?” (Appendix A).

### 2.6. Independent Variables

The analysis considered various independent variables, encompassing socio-demographic factors such as age, race, ethnicity, and family income classified based on federal poverty level. TSE was assessed based on two questions. The first question determined if anyone in the household used cigarettes, cigars, or pipe tobacco, with coding as 0 for No and 1 for Yes. The second question assessed whether anyone smoked inside the child’s home, with coding options “No one smokes in the household”, “Someone smokes, not inside the house”, and “Someone smokes inside the house”.

### 2.7. Data Analysis

Statistical analyses were carried out using Stata version 17.0 [33]. Continuous variables were expressed as mean ± standard deviation, and categorical variables were presented as frequency (percentage). Initial comparisons between the two groups (children with and without TSE) in terms of continuous variables employed t-test. Chi Squares were employed for comparing categorical variables. A logistic regression model for univariate and multiple independent variables was employed to investigate the connection between neuropsychiatric co-morbidities, socio-demographic factors, and TSE. In the univariate models, each covariate’s association with TSE was independently assessed. The multivariate models examined the association between each covariate and TSE while adjusting for all other covariates. The regression models generated adjusted odds ratios (ORs) and 95% confidence intervals (CIs).

## 3. Results

The dataset for the years 2020–2021 from NSCH consisted of 93,669 participants, including children and adolescents aged 0 to 17 years. Within this group, 91,404 individuals aged 0 to 17 years responded to questions regarding the TSE and were the focus of this study. We included 91,404 children and adolescents aged between 0 to 17 years for the TSE prevalence and socio-demographic analysis, and 79,182 children and adolescents aged between 3 to 17 years for the neuropsychiatric co-morbidities analysis. The participants had an average age of 8.7 ± 5.3 (mean ± SD). Significantly, 11,751 individuals, accounting for 12.9%, had confirmed exposure to tobacco smoke (Table 1).

The multivariate analysis revealed notable increases in the probability of TSE in the adolescent age group (11–17 year-olds) (odds ratio (OR): 1.2, *p*-value < 0.001), the multi-race ethnic group (OR: 1.1, *p*-value < 0.01). In contrast, being female (OR: 0.9, *p*-value: 0.001), having a higher household income (OR: 0.3, *p*-value < 0.001), and the Hispanic (OR: 0.6, *p*-value < 0.001), Asian (OR: 0.7, *p*-value < 0.001), and Black (OR: 0.6, *p*-value < 0.001) ethnic groups were associated with reduced odds of TSE among youths (Table 2).

A total of 36.4% of youths with TSE had at least one neuropsychiatric co-morbid condition. The most common neuropsychiatric co-morbid condition with TSE was anxiety problems (15.7%), followed by ADHD (15.5%), behavioral and conduct problems (13.7%), and learning disability (12%), respectively (Figure 1).

The odds of co-morbid anxiety problems (OR: 0.3, *p*-value: 0.02) and ASD (OR: 0.9, *p*-value: 0.04) were significantly lower in females compared to males (Table 3). Regarding ethnic differences, the odds of co-morbid ADHD (OR: 0.3, *p*-value: 0.001), anxiety problems (OR: 0.4, *p*-value: 0.003), speech/other language disorder (OR: 0.4, *p*-value: 0.001), developmental delay (OR: 0.4, *p*-value: 0.001), behavioral and conduct problems (OR: 0.4, *p*-value: 0.003), and learning disability (OR: 0.5, *p*-value: 0.004) were lower among individuals of Asian ethnicity (Table 3). Conversely, American Indian children and adolescents had higher odds of co-morbid headaches (OR: 3, *p*-value: 0.005) (Table 3).

In terms of the severity of co-morbid conditions, our findings indicate that the co-occurrence of TSE with TS (Tourette’s syndrome) (OR: 4.4, *p*-value < 0.001), ADHD (OR: 1.3, *p*-value < 0.001), developmental delay (OR: 1.3, *p*-value < 0.001), behavioral and conduct problems (OR: 1.3, *p*-value < 0.001), frequent/severe headaches (OR: 1.3, *p*-value: 0.005), depression (OR: 1.2, *p*-value: 0.02), anxiety problems (OR: 1.2, *p*-value< 0.01), ASD (OR: 1.2, *p*-value < 0.001), and learning disability (OR: 1.2, *p*-value: 0.03) may contribute to a more severe manifestation of these neuropsychiatric conditions (Table 3).

## 4. Discussion

We found higher prevalence rates of TSE in males, school-aged adolescents, and specific racial groups. Additionally, our findings underscored the prevalence of co-occurring conditions such as anxiety, ADHD, behavioral/conduct problems, and learning disabilities among youths with TSE, emphasizing the detrimental effects of TSE on neuropsychiatric health. We observed that the co-occurrence of TSE with various conditions leads to increased severity, aligning with previous research indicating TSE’s adverse effects on neuropsychiatric development.

### 4.1. Prevalence and Sociodemographic Characteristics of TSE

We discovered that 11,751 individuals, constituting 12.9% of our sample, were exposed to tobacco smoke. This exposure was notably higher among males and among age groups encompassing school-age youths and adolescence (6–17 years old). The prevalence we observed is lower than that documented in studies [34]. Merianos et al. examined the 2018–2019 NSCH dataset and evaluated tobacco smoke exposure among children aged 6 to 11 years [35]. They identified a TSE prevalence of 14.6%. The racial breakdown mirrors a comparable pattern, with most TSE cases occurring among non-Hispanic whites. However, our research diverges in noting a higher odds ratio for American Indian/Alaska Native (OR: 1.7) and multiracial individuals (OR: 1.2), as opposed to the 2018–2019 data, which indicate a greater proportion of TSE among the Hispanic population [35]. Likewise, data from the Centers for Disease Control and Prevention (CDC), sourced from the National Health and Nutrition Examination Survey, revealed that tobacco smoke exposure was more prevalent among younger (aged 3–11) compared to older (aged 12–17) youths, aligning with our own discoveries [36]. They observed similar exposure rates between boys and girls, which contrasts with our findings indicating significantly higher exposure among males. Additionally, they noted that non-Hispanic black youths exhibited the highest exposure rate (61.8%), followed by non-Hispanic white (34.3%), Hispanic (24.9%), and non-Hispanic Asian (18.3%) [1].

The tobacco exposure rates from the 2013–2014 US National Health and Nutrition Examination Survey (NHANES) mirror sociodemographic patterns similar to those observed in our study. Their findings reveal that among male youths aged 6–11, the prevalence of tobacco smoke exposure was higher compared to female youths aged 3–5. Additionally, non-Hispanic black children were identified as 1.85 times more likely to be exposed than non-Hispanic white children [24]. In the randomized controlled trial conducted by Dempsey et al. (2015), aimed at discerning racial disparities in tobacco exposure between non-Hispanic black and non-Hispanic white populations, 17,692 parents were screened within a pediatric setting. The study revealed a lower prevalence of smoking among black parents compared to white parents. Notably, a significant limitation highlighted in the study was the observation of bias, wherein black parents were more frequently questioned about cigarette smoking than their white counterparts. The authors of the study postulate that this bias might stem from an erroneous association of black parents with lower socioeconomic status (SES), possibly influenced by US Census data indicating higher instances of low SES among black families [37].

A national study examining US population demographic trends spanning from 1975 to 2015 reflects similar ethnic disparities in tobacco usage among families, aligning closely with the findings of our research [38]. Notably, this study also suggests a correlation between greater tobacco use and lower socioeconomic status. Drawing from the preceding NSCH dataset (2018–2019), it becomes evident that families with lower household incomes, particularly those falling within the 0–199% federal poverty level, demonstrate markedly higher incidences of tobacco smoke exposure (TSE) at 54.8%, in contrast to those within the 200–399% federal poverty level (27.2%) [39]. This phenomenon likely stems not only from a lack of awareness but also from the pervasive stress experienced within neighborhood and work environments, underscoring the imperative for tailored considerations when formulating tobacco control policies for vulnerable groups [40,41,42]. This is likely not only due to a lack of awareness but also due to overall stress in neighborhood and work environments, emphasizing the need for additional considerations for susceptible groups when implementing tobacco control policies [43]. Moran et al. analyzed 2013–2014 PATH data, revealing higher tobacco marketing in lower socioeconomic groups. They argue that “communication inequalities” contribute to health disparities by underexposing disadvantaged communities to quality health communication and overexposing them to negative health influences like tobacco marketing [44].

Based on our findings, overexposure to negative health influences in low socio-demographic populations should be a focus in future health policies. Additionally, teaching healthy stress coping mechanisms that do not involve substance use requires a multidisciplinary approach involving physicians, therapists, and community support [45]. Implementing additional training for health care professionals on tobacco cessation and education for patients may lead to better smoking cessation rates, especially for individuals who are of lower socio-demographic status [46,47].

### 4.2. Co-Morbid Conditions and Their Sociodemographic Risk Factors

We found that anxiety, ADHD, behavioral/conduct problems, and learning disabilities were the most prevalent co-occurring conditions with TSE. Our findings show that 36.4% of youths with TSE had at least one neuropsychiatric comorbidity. We observed that the odds of co-morbid anxiety problems and ASD were significantly lower in females compared to males. These results contrast with the previous literature on males and anxiety, which noted higher anxiety rates in hypogonadism males [48]. A 2022 systematic review highlights the limited exploration of psychosocial and biological factors together in gender-based anxiety research [49]. While femininity is often linked to higher anxiety, caution is needed due to potential biases in data collection and the impact of gender norms. Females displaying traditionally masculine traits show lower anxiety, suggesting the influence of personality over biological sex [50]. Similarly, this may be accounting for the differences in gender observed in our study results.

We observed significantly lower odds of ASD as a co-occurring condition in females, possibly influenced by perceived gender norms affecting the accurate diagnosis of girls with ASD. This is notable when considering themes such as gendered symptoms, behavioral manifestations, relational dynamics, social communication, language skills, and restricted or repetitive behaviors [22]. Our study, utilizing surveys and various communication methods, did not delve into the intricate diagnostic processes of comorbid conditions or their potential heterogeneity.

When comparing comorbidities across race and ethnicity, Asians exhibited decreased odds of comorbid ADHD, anxiety, speech or language disorders, developmental delay, and behavioral and conduct problems comparing to Caucasians and Caucasians had highest base line odds for all of the comorbid conditions. Similar to these findings, Bandiera’s 2011 study on U.S. children aged 8–15 with second-hand smoke exposure, exhibited higher symptoms of MDD, GAD, ADHD, and CD. The study emphasized that the associations were higher in boys and non-Hispanic white participants [51]. These results indicate a notable impact of culture on the data collected in our sample population surveys and the diagnostic process for comorbid conditions. Cultural influences, especially variations in perceptions of challenging behaviors, may contribute to these findings [31].

Our results highlight the negative impact of TSE on neuropsychiatric development. This emphasizes the crucial need for culturally and demographically sensitive clinical tools to diagnose developmental abnormalities. Early and equitable intervention is essential to address the co-occurring disorders prevalent among America’s diverse youths exposed to tobacco smoke.

### 4.3. Effects of TSE on the Severity of Co-Morbid Conditions

We found that the co-occurrence of TSE with TS, developmental delay, behavioral and conduct problems, headaches, depression, anxiety problems, ASD, and learning disability is associated with a more severe manifestation of these neuropsychiatric conditions. These results aligns with the previous findings that TSE has been associated with an increased severity of co-morbid conditions [52,53,54,55].

Although this study does not use prenatal data, TSE during pregnancy poses risks such as sudden infant death syndrome, low birth weight, respiratory issues, and metabolic syndrome [56]. The brain is also more vulnerable during developmental periods like gestation and adolescence, susceptible to harmful effects from nicotine use [57]. These effects involve complex mechanisms, including direct neurotoxicity from tobacco constituents and indirect factors like prenatal exposure and environmental influences [57]. Nicotine can also disrupt neurotransmitter systems crucial for neurodevelopment [55]. Prenatal nicotine exposure (PNE) can lead to impulsivity and disrupt neural activity in the prefrontal cortex, potentially contributing to ADHD symptoms [58]. Children exposed to tobacco smoke face increased health risks and exhibit difficulties in executive functions, attention deficits, and hyperactive behavior [59]. Moreover, PNE is associated with a higher risk of severe neuropsychiatric disorders in offspring, especially ADHD with comorbidities [60,61]. Similarly, studies show strong links between maternal smoking and TS, especially with ADHD [62,63], and avoiding maternal smoking may lessen TS symptoms [64].

Similarly, in ASD, children and adolescents exposed to tobacco smoke may experience a worsening of behavioral symptoms and impaired social interactions [65]. Maternal smoking during first and third trimesters is also associated with elevated risks of conduct disorder symptoms in offspring. The risk increases with the level of maternal smoking [4]. The Family Life Project’s study on 1096 children shows a clear link between early environmental smoke exposure, measured by salivary cotinine, and later hyperactivity and conduct problems. The findings underscore the need to reduce children’s environmental smoke exposure beyond prenatal exposure from parental smoking [66].

The potential link between PNE and mood disorders in offspring involves harmful tobacco smoke compounds crossing the placenta, impacting the developing brain and disrupting neurodevelopmental pathways [67]. PNE is suggested to induce epigenetic changes, affecting genes linked to the hypothalamic-pituitary-adrenocortical (HPA) axis and potentially influencing mood disorder development in offspring [68]. A study using ALSPAC data suggests a potential link between PNE and hypomania in young adulthood. While evidence for maternal smoking during pregnancy and lifelong hypomania is weak, a strong association was found in individuals with both hypomania and psychotic symptoms, indicating that in utero smoking exposure may increase the risk of more severe psychopathology on the mood-psychosis spectrum [69]. Adolescent nicotine exposure can also lead to long-term dysregulation of mesocorticolimbic states, altering sensitivity of serotonergic and dopaminergic receptors, as well as brain-derived neurotrophic factor (BDNF) in the striatum and cortex. This dysregulation may impair neural growth and circuit formation, contributing to behavioral abnormalities and mood disorders [70,71].

Tobacco smoke also induces oxidative stress and systemic inflammation, with high cytokine levels associated with anxiety and depressive mood [72,73]. Eunmi Lee et al. conducted a study based on the 2018 Korea youths risk behavior web-based survey, with 51,500 students, which revealed a positive correlation between increasing second hand smoke exposure and higher risks of stress, depression, and suicidal ideation [74].

Early risk management for TSE needs a multifaceted approach to target harm reduction and minimize health risks. Education and awareness, targeting vulnerable groups, are crucial via where public campaigns to inform individuals about the dangers of both smoking and second-hand smoke. Legislation and policy play an important role in smoke-free laws in public spaces, workplaces, and multi-unit housing, while higher tobacco taxes can discourage smoking. Resources, counseling, and medication to assist individuals in quitting smoking may also be helpful. Lastly, environmental controls, like effective ventilation systems and designated smoking areas, may help manage smoke exposure, and ongoing surveillance is important for assessing tobacco smoke levels, the effectiveness of policies, and community engagement through local initiatives. Clinicians should integrate this knowledge into their screening and treatment plans, focusing on smoking cessation initiatives to improve both maternal and child mental health outcomes. Understanding the impact of tobacco exposure in youths on increased neuropsychiatric severity is crucial for tailoring early interventions, improving clinical assessments, guiding treatment strategies, informing public health initiatives, empowering parental education, and advancing research for enhanced understanding and targeted interventions.

## 5. Limitations

The major limitation of this study is the lack of a valid diagnostic instrument and parent-based reports, which can be a source of bias. NSCH is cross-sectional in nature and causality cannot be ascertained. Moreover, caregivers were not questioned about prenatal TSE, leaving the potential impact of in utero exposure uncertain, frequency, duration, and amount of TSE, which could have influenced observed associations. Additionally, other settings for TSE, such as inside vehicles and during home care, as well as various forms of TSE (e.g., cigars, electronic cigarettes), and differing levels of exposure were not assessed, potentially resulting in underestimated exposure. Furthermore, caregiver-reported diagnoses of their child’s mental health and neurobehavioral conditions, as well as the severity of these conditions, may have been influenced by recall bias or the caregivers’ perception of their child’s condition. We did not test for the normality of continuous variables in the data, which may be worth exploring in future studies.

## 6. Conclusions

In conclusion, our findings suggest that TSE is significantly associated with gender, race, and household income. The prevalent co-morbid conditions with TSE included anxiety problems, ADHD, behavioral and conduct problems, and learning disability. Additionally, TSE can influence the severity of these co-morbid neuropsychiatric conditions. Identifying and addressing these co-morbidities early is crucial for clinical management and support.

## Figures and Tables

**Figure 1 healthcare-12-02102-f001:**
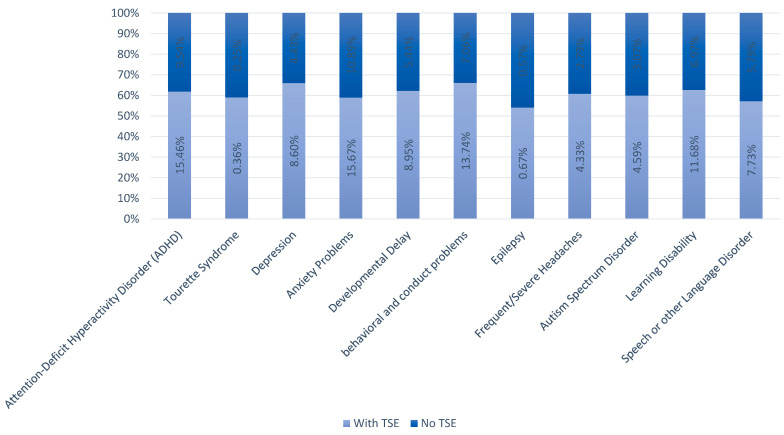
Prevalence of co-morbid conditions with and without TSE.

**Table 1 healthcare-12-02102-t001:** Socio-demographic characteristics of the tobacco smoke exposure (TSE).

Socio-Demographic Characteristics		Total (n = 91,404)	With TSE (n = 11,751)	Without TSE (n= 79,653)
**Age (Years)**		**8.7 ± 5.3**	**9.3 ± 5.1**	**8.6 ± 5.2**
Age Groups	Preschool: 3–5	18,306 (23.6%)	2176 (21%)	16,130 (24%)
	School: 6–10	21,843 (28.1%)	2853 (27.4%)	18,990 (28.2%)
	Adolescents: 11–17	37,473 (48.3%)	5368 (52%)	32,105 (48%)
Sex	Male	47,405 (52%)	6234 (53%)	41,171 (52%)
	Female	43,999 (48.1%)	5517 (47%)	38,482 (48.3%)
Race	White	60,570 (66.3%)	7889 (67.1%)	52,681 (66.1%)
	Hispanic	12,321 (13.5%)	1399 (12%)	10,922 (13.7%)
	Black	5930 (6.5%)	824 (7%)	5106 (6.4%)
	Asian	5058 (5.5%)	436 (3.7%)	436 (5.8%)
	American Indian/Alaska Native	572 (0.6%)	115 (1%)	457 (0.6%)
	Native Hawaiian/Other Pacific Islander	271 (0.3%)	46 (0.4%)	225 (0.3%)
	Multi-Race	6682 (7.3%)	1042 (8.9%)	5640 (7%)
Federal Poverty Level	≥400%	37,176 (40.7%)	2546 (21.7%)	34,630 (43.5%)
	200%–399%	27,973 (31%)	3830 (32.6%)	24,143 (30.3%)
	100%–199%	14,905 (16.3%)	2832 (24.1%)	12,073 (15.1%)
	<100%	11,350 (12.4%)	2543 (21.6%)	34,630 (43.5%)

**Table 2 healthcare-12-02102-t002:** Predictors of tobacco smoke exposure (TSE) based on univariate and multivariate analysis.

Socio-Demographic Predictors		Univariate Analysis OR (95% CI)	Multivariate Analysis OR (95% CI)
Age Groups	Preschool: 3–5	Reference	Reference
	School: 6–10	1.1 (1.1.1) ***	1.1 (1–1.1) *
	Adolescents: 11–17	1.2 (1.2–1.3) ***	1.2 (1.2–1.3) ***
Sex	Male	Reference	Reference
	Female	0.9 (0.2–0.3) **	0.9 (0.9–0.98) *
Race	White	Reference	Reference
	Hispanic	0.9 (0.8–0.9) ***	0.6 (0.5–0.6) ***
	Black	1.07 (0.9–1.1)	0.7 (0.6–0.7) ***
	Asian	0.6 (0.6–0.7) ***	0.6 (0.5–0.6) ***
	American Indian/Alaska Native	1.7 (1.4–2.1) ***	1.1 (0.9–1.4)
	Native Hawaiian/Other Pacific Islander	1.4 (0.9–1.8)	0.9 (0.7–1.3)
	Multi-Race	1.2 (1.1–1.3) ***	1.1 (1.1–1.2) **
Federal Poverty Level	<100%	Reference	Reference
	100%–199%	0.8 (0.8–0.9) ***	0.9 (0.8–0.9) ***
	200%–399%	0.5 (0.5–0.6) ***	0.6 (0.6–0.7) ***
	≥400%	0.2 (0.2–0.3) ***	0.3 (0.2–0.3) ***

*: *p* value < 0.05, **: *p* value < 0.01, ***: *p* value < 0.001.

**Table 3 healthcare-12-02102-t003:** Neuropsychiatric co-morbidities in tobacco smoke exposure (TSE).

Disorders	TSE (n = 11,751)	Sex OR (95% CI)	Severity OR (95% CI)	Race
Attention-Deficit Hyperactivity Disorder (ADHD)	1596 (15.5%) ***	M: Reference F: 0.9 (0.9–1.1)	1: Reference 2: 1.3 (1.2–1.5) ***	White: Reference Hispanic: 0.9 (0.7–1.04) Black: 0.9 (0.7–1.1) Asian: 0.3 (0.1–0.5) *** American Indian: 1 (0.5–2.1) Native Hawaiian: 1.5 (0.5–5) Multi-Race: 1.1 (0.9–1.3)
Tourette Syndrome (TS)	37 (0.36%) *	M: Reference F: 1.2 (0.5–2.6)	1: Reference 2: 4.4 (2.1–9.4) ***	White: Reference Hispanic: 1.3 (0.5–3.4) Black: 0.4 (0.04–3.1) Asian: 1 (0.1–1.6) American Indian: 4.2 (0.2–70) Native Hawaiian: 1 (0.2–57) Multi-Race: 1 (0.9–11)
Depression	888 (8.6%) ***	M: Reference F: 0.9 (0.8–1)	1: Reference 2: 1.2 (1–1.4) *	White: Reference Hispanic: 0.8 (0.6–1) Black: 1.1 (0.8–1.5) Asian: 0.6 (0.3–1.1) American Indian: 1.3 (0.6–2.6) Native Hawaiian: 0.5 (0.1–4.5) Multi-Race: 1.1 (0.8–1.4)
Anxiety Problems	1614 (15.7%) ***	M: Reference F: 0.35 (0.3–0.4) *	1: Reference 2: 1.2 (1.1–1.4) ***	White: Reference Hispanic: 0.9 (0.8–1.1) Black: 1.2 (0.9–1.5) Asian: 0.4 (0.2–0.7) ** American Indian: 1.1 (0.6–1.9) Native Hawaiian: 0.4 (0.05–3) Multi-Race: 1.2 (0.9–1.4)
Autism Spectrum Disorder	475 (5%) ***	M: Reference F: 0.9(0.7–0.9) *	1: Reference 2: 1.2 (1–1.5) ***	White: Reference Hispanic: 0.6 (1.1–2) ** Black: 0.7 (0.5–1.1) Asian: 0.6 (0.3–1.1) American Indian: 0.9 (0.2–3.4) Native Hawaiian: 2.5 (0.4–15) Multi-Race: 0.7 (0.5–1.1)
Developmental Delay	926 (9%) ***	M: Reference F: 1 (0.8–1.1)	1: Reference 2: 1.3 (1.1–1.5) ***	White: Reference Hispanic: 0.7 (0.5–0.8) *** Black: 0.8 (0.6–1) Asian: 0.4 (0.3–0.7) ** American Indian: 0.7 (0.3–1.6) Native Hawaiian: 0.6 (0.6–1) Multi-Race: 0.8 (0.6–1.03)
Learning Disability	1208 (12%) ***	M: Reference F: 0.9 (0.7–1.01)	1: Reference 2: 1.2 (1–1.3) *	White: Reference Hispanic: 0.7 (0.6–0.9) ** Black: 0.9 (0.7–1.1) Asian: 0.5 (0.3–0.8) ** American Indian: 1 (0.5–1.9) Native Hawaiian: 1.2 (0.4–3.7) Multi-Race: 0.9 (0.7–1.2)
Behavioral and Conduct Problems	1419 (13.7%) ***	M: Reference F: 1 (0.9–1.2)	1: Reference 2: 1.3 (1.1–1.4) ***	White: Reference Hispanic: 0.6 (0.5–0.8) *** Black: 0.9 (0.7–1.1) Asian: 0.4 (0.2–0.7) ** American Indian: 0.9 (0.9–1.8) Native Hawaiian: 1.4 (0.4–4.4) Multi-Race: 0.9 (0.7–1.1)
Speech or other Language Disorder	799 (7.7%) ***	M: Reference F: 0.9 (0.8–1.1)	1: Reference 2: 1.04 (0.9–1.2)	White: Reference Hispanic: 0.8 (0.7–1.1) Black: 0.8 (0.6–1.1) Asian: 0.4 (0.2–0.7) ** American Indian: 0.7 (0.3–1.7) Native Hawaiian: 1 (0.7–16) Multi-Race: 0.8 (0.6–1.1)
Seizure Disorder or Epilepsy	78 (0.67%)	M: Reference F: 0.8 (0.5–1.4)	1: Reference 2: 1 (0.7–1.8)	White: Reference Hispanic: 1.2 (0.6–2.3) Black: 1 (0.4–2.5) Asian: 0.3 (0.04–2.6) American Indian: 1.4 (0.1–13) Native Hawaiian: 1 (0.5–7) Multi-Race: 0.7 (0.3–1.8)
Frequent/Severe Headaches	448 (4.3%) ***	M: Reference F: 0.9 (0.7–1)	1: Reference 2: 1.3 (1–1.6) *	White: Reference Hispanic: 1 (0.7–1.4) Black: 1.2 (0.8–1.8) Asian: 0.3 (0.07–1.2) American Indian: 3 (1.3–7.1) ** Native Hawaiian: 1 (0.8–8.7) Multi-Race: 1.3 (0.8–1.8)

*: *p* value < 0.05, **: *p* value < 0.01, ***: *p* value < 0.001. M: male, F: female, 1: mild, 2: moderate to severe.

## Data Availability

This is an analysis of a preexisting publicly available anonymized dataset. No new patient data was created in our study. The dataset can be found on https://www.childhealthdata.org/learn-about-the-nsch/NSCH (accessed on 30 June 2024).

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
