# Peer review of "Tobacco Smoke Exposure in Children and Adolescents: Prevalence, Risk Factors and Co-Morbid Neuropsychiatric Conditions in a US Nationwide Study"

_healthcare, 2024, doi:10.3390/healthcare12212102_

Round 1
Reviewer 1 Report
Comments and Suggestions for Authors
The topis of this paper is tobacco smoke exposure, particularly in children and adolescents, and its association with neuropsychiatric conditions. It is a major public health concern that needs to be discussed more. The strength of the study is its design and a good quality dataset with many observations. What the paper would benefit from and needs improvement on is a proper description of data analysis, better presentation of results, and some additional calculations (and possibly some changes in the analysis) as I outline below.
Abstract could be more concise.
The aim doesn't include risk factors.
- line 29: abbreviation ASD used but not defined
- line 48: I believe it should be "health of non-smokers" instead of "health of smokers"
- line 116: "The nature of the questions varied according to the age of the selected child." Please provide whole surveys as supplementary material.
- line 119: Suppl. Table 1 doesn't include questions about topics listed in the text, eg parental health. All survey questions should be provided.
Regarding data analysis and presentation of results:
How did you test for normality of continuous variables?
What are Chi Squares and why did you use them for continuous variables?
What kind of regression models did you build (I assume logistic regression but it is not stated explicitly)?
The term multivariate (multiple outcomes) is used incorrectly, I believe you meant multivariable (multiple independent variables) - see https://doi.org/10.1093/ntr/ntaa055 for example.
Line 174: "indicating the increased odds of TSE associated with each covariate after accounting for other variables." - this is an inappropriate statement in the description of data analysis, how do you know at this moment that the odds are increased? Also, it is a repetition of "while adjusting for all other covariates" from line 172.
Line 185: the term likelihood is used incorrectly. Assuming logistic regression was used, these models produce probabilities and odds ratios, but not likelihood. Although probability and likelihood are synonyms in layman's terms, from a statistical perspective they are distinct concepts.
Line 177 - "individuals aged 0 to 17 years responded" individuals or their caregivers as stated in section 2.2?
Tables 1 & 2 - in the text you state that you included individuals aged 0-17 for socio-demographic characteristics, however, in these Tables you show data for ages 3-17. Also, the total for age groups (18,306+21,843+37,473=77,622) does not agree with the total from the 1st row (91,404). I did not check other totals, please double-check the numbers. The result of univariate analysis for age group 6-10 is "1.1 (1.1.1)***" - missing hyphen in the CI.
Reporting odds ratios - please ensure a consistent use of the same number of significant digits. In the text, I would inlcude confidence intervals instead of p-values as the width of the interval gives more practical information than p value.
The information from Figure 1 would be better presented in a table. The range of y-axis and the percentages on the bars are misleading.
Headings of tables - please provide short descriptions of what you present in the tables.
I think Tables 1 & 2 could be connected into one table.
Regarding multivariable models, instead of building the model on all available variables, choose the ones that you specifically want to adjust for.
Line 212: using the verb "contribute" suggests causality, and this cannot be concluded from this study as you point out in the Limitations section. I suggest using phrases such as "is associated", "increases/decreases the odds" etc instead.
I would perform additional calculations to compare the prevalence and severity of neuropsychiatric disorders between TSE and non-TSE groups. This would provide more comprehensive insight into the available data. In line 392 you wrote "TSE can influence the severity of these co-morbid neuropsychiatric conditions" - how did you come to this conclusion without examining the severity in the non-TSE group?
After improving data analysis, please modify the conclusion section accordingly.
Please add another limitation: frequency (how many times per day) and duration (months or years) of TSE could not be quantified in this study. We can hypothesize that greater frequency and/or duration can be associated with greater prevalence and/or severity of neuropsychiatric disorders.
Author Response
Thank you for your review and for providing detailed feedback. Please find below the responses in italics and the changes made to the revised file submitted with MDPI, survey questions are attached in word doc:
The topics of this paper are tobacco smoke exposure, particularly in children and adolescents, and its association with neuropsychiatric conditions. It is a significant public health concern that needs to be discussed more. The strength of the study is its design and a good quality dataset with many observations. What the paper would benefit from and needs improvement on is a proper description of data analysis, better presentation of results, and some additional calculations (and possibly some changes in the analysis) as I outline below.
Abstract could be more concise: Attempted to make it more concise.
The aim doesn't include risk factors.
- Added line about sociodemographic factors in aim.
- line 29: abbreviation ASD used but not defined
Defined in line 28
- line 48: I believe it should be "health of non-smokers" instead of "health of smokers."
Added: changed "health of smokers" to "health of non-smokers."
- line 116: "The nature of the questions varied according to the age of the selected child." Please provide whole surveys as supplementary material.
The information we provided in the methods part is about the whole dataset. You can find all of the survey questions in this link: 2021 NSCH Guide to Topics and Questions (childhealthdata.org), However the questions that were used in this study is provided in the supplemental table 1.
Line 119: Suppl. Table 1 doesn't include questions about topics listed in the text, such as parental health. All survey questions should be provided.
We provided the relevant survey questions. A link to all the survey questions has been added.
Regarding data analysis and presentation of results:
How did you test for normality of continuous variables?
- We did not test for normality of continuous variables in this paper, we have added this as the limitation in our paper and would be worth considering in future studies.
What are Chi Squares and why did you use them for continuous variables?
- Chi Sqaure was used for categorical variables, while t-test was used for comparing continuous variables-clarified in methodology
What kind of regression models did you build (I assume logistic regression but it is not stated explicitly)?
- We used logistic regression, now stated explicitly in the methods section in attached revision.
The term multivariate (multiple outcomes) is used incorrectly, I believe you meant multivariable (multiple independent variables) - see https://doi.org/10.1093/ntr/ntaa055 for example.
- Changed "multivariate" to "multiple independent variables"
Line 174: "indicating the increased odds of TSE associated with each covariate after accounting for other variables." - this is an inappropriate statement in the description of data analysis, how do you know at this moment that the odds are increased? Also, it is a repetition of "while adjusting for all other covariates" from line 172.
- Deleted the statement in revised version
Line 185: the term likelihood is used incorrectly. Assuming logistic regression was used, these models produce probabilities and odds ratios, but not likelihood. Although probability and likelihood are synonyms in layman's terms, from a statistical perspective they are distinct concepts.
- "Likelihood" changed to "probability" in results.
Line 177 - "individuals aged 0 to 17 years responded" individuals or their caregivers as stated in section 2.2?
Tables 1 & 2 - in the text you state that you included individuals aged 0-17 for socio-demographic characteristics, however, in these Tables you show data for ages 3-17.
We included 91,404 children and adolescents aged between 0 to 17 years for the TSE prevalence and socio-demographic analysis and 79,182 children and adolescents aged between 3 to 17 years for the neuropsychiatric co-morbidities analysis.
Also, the total for age groups (18,306+21,843+37,473=77,622) does not agree with the total from the 1st row (91,404). I did not check other totals, please double-check the numbers.
We included the 3 age groups
This discrepancy arises because we initially only included the three age groups (3-17 years old) in our analysis, as we only had data on comorbid psychiatric conditions for these age groups. To maintain consistency between the prevalence and comorbidity analyses, we kept the same age groups. However, for the sake of completeness, we have now also included data for the under-3-year-old age group
The result of univariate analysis for age group 6-10 is "1.1 (1.1.1)***" - missing hyphen in the CI.
Thank you for noting. It is corrected.
Reporting odds ratios, please ensure a consistent use of the same number of significant digits. In the text, I would include confidence intervals instead of p-values, as the width of the interval gives more practical information than the p value.
- Corrected the numerical inconsistencies
The information from Figure 1 would be better presented in a table. The range of y-axis and the percentages on the bars are misleading.
This data is also presented in Table 3.
Headings of tables - please provide short descriptions of what you present in the tables.
I think Tables 1 & 2 could be connected into one table
Headers provide descriptions of the results discussed, however presenting tables 1 and 2 could become confusing for the readers- changed headings for clarity
Regarding multivariable models, instead of building the model on all available variables, choose the ones that you specifically want to adjust for.
We only adjusted the model based on age, race, sex, and poverty level which all affect the TSE prevalence.
Line 212: using the verb "contribute" suggests causality, and this cannot be concluded from this study as you point out in the Limitations section. I suggest using phrases such as "is associated ", "increases/decreases the odds" etc instead.
- Changed "contribute" to "is associated"
I would perform additional calculations to compare the prevalence and severity of neuropsychiatric disorders between TSE and non-TSE groups. This would provide more comprehensive insight into the available data. In line 392 you wrote "TSE can influence the severity of these co-morbid neuropsychiatric conditions" - how did you come to this conclusion without examining the severity in the non-TSE group?
All of the comparisons for the comorbidities were between the two groups of kids who had the comorbid condition and TSE and the group who had the comorbid condition but did not have TSE ( between non TES and TSE).
After improving data analysis, please modify the conclusion section accordingly.
All of the comparisons for the comorbidities were between the two groups of kids who had the comorbid condition and TSE and the group who had the comorbid condition but did not have TSE ( between non TES and TSE).
Please add another limitation: frequency (how many times per day) and duration (months or years) of TSE could not be quantified in this study. We can hypothesize that greater frequency and/or duration can be associated with greater prevalence and/or severity of neuropsychiatric disorders.
Added the limitation for frequency and duration of TSE into limitations

Reviewer 2 Report
Comments and Suggestions for Authors
Dear Authors,
thank you for your submission. The manuscript is very interesting considering the implications in the clinical practice. moreover, it is well written and organized. However, some little changes are needed.
Do Authors have any information regarding the use of substances in this sample (both parents and children)? This would be interesting in order to understand the possibility of a familiar risk. Please modify the Results according to this data and add the following refs:
https://pubmed.ncbi.nlm.nih.gov/34025478/
https://pubmed.ncbi.nlm.nih.gov/34204131/
https://pubmed.ncbi.nlm.nih.gov/34886357/
In the Discussion there is no mention regarding early prevention interventions on family at risk. Please add a possible approach to be used.
Author Response
Thank you for providing feedback and reviewing our work. Please find responses in italics and changes made to the revised version of the document submitted to MDPI:
Dear Authors,
thank you for your submission. The manuscript is very interesting considering the implications in the clinical practice. moreover, it is well written and organized. However, some little changes are needed.
Do Authors have any information regarding the use of substances in this sample (both parents and children)? This would be interesting in order to understand the possibility of a familiar risk. Please modify the Results according to this data and add the following refs:
https://pubmed.ncbi.nlm.nih.gov/34025478/
https://pubmed.ncbi.nlm.nih.gov/34204131/
https://pubmed.ncbi.nlm.nih.gov/34886357/
Thank you for the feedback. In this study, we exclusively discuss the risk of Tobacco smoke exposure to nonsmokers and, unfortunately, did not consider the use of substances other than tobacco.
In the Discussion there is no mention regarding early prevention interventions on family at risk. Please add a possible approach to be used.
Added in the last paragraph
Reviewer 3 Report
Comments and Suggestions for Authors
This is an epidemiological papers that is looking at associations of Tobacco smoke exposure with mental health conditions. They use data from 2020-21 to showcase their findings.
Introduction - the introduction is coming across biased and leave no room to offer other explanations. Generally in majority of the writing the authors do not review other explanations about mental health conditions. In addition they take two elements and correlate it with each other and as premise this should be treated more cautiously, considering this is correlational.
Method: Why are you using only 2020-21 data? Wouldn't that argument be more stronger if you could show a longer trend? In addition, do you take into account other variables, e.g. smoking of marijuana, air pollution? what about genetic factors?
Results: Did you control for any other variables e.g. SES? In your discussion you refer to results/analyses that are not reported.
Discussion:
Line 322- 326: Could you clarify where you report the results about co-occurence? Currently I cannot find any reference in the results for this.
Line 327-340 - The authors write about TSE in pregnancy but have not reported data and themselves state that this was not asked. This section as such it is speculative and not supported by evidence.
Line 364-368 - IT is not clear what point the authors are trying to make with that paragraph. Could you elaborate more?
Line 369-375 - It is good that you are considering clinical implications, but it is not clear how you would expect someone to integrate, could you provide examples of what that would look like? Any successful examples/citations of what you are suggesting? This section comes across as an afterthought and for someone that is working in behaviour change and developing interventions this section is not adding something that is not already known.
Limitation Section. your limitation section does not take into account some key methodological limitations of your research. Please review.
Author Response
Thank you for reviewing our work and providing your feedback. Please find the responses below in italics and the changes made in the revised document submitted with MDPI:
This is an epidemiological papers that is looking at associations of Tobacco smoke exposure with mental health conditions. They use data from 2020-21 to showcase their findings.
Introduction - the introduction is coming across biased and leave no room to offer other explanations. Generally in majority of the writing the authors do not review other explanations about mental health conditions. In addition they take two elements and correlate it with each other and as premise this should be treated more cautiously, considering this is correlational.
Made changes to the introduction appropriately
Method: Why are you using only 2020-21 data? Wouldn't that argument be more stronger if you could show a longer trend? In addition, do you take into account other variables, e.g. smoking of marijuana, air pollution? what about genetic factors?
The NSCH dataset is not a longitudinal dataset so we could not follow trends based on that. Also we only had access to the NSCH 20-21 at the time of performing this study. The NSCH dataset does not provide variables related to air pollution, Marijuanna smoking or genetic factors. We added those to the limitation.
Results: Did you control for any other variables e.g. SES? In your discussion you refer to results/analyses that are not reported.
Added pointers in discussion quoting our results section. We reported the rates related to the Federal poverty level of the families.
Discussion:
Line 322- 326: Could you clarify where you report the results about co-occurence? Currently I cannot find any reference in the results for this.
Line 220- co-occurance of psychiatric comorbidities, also please refer table 3
Line 327-340 - The authors write about TSE in pregnancy but have not reported data and themselves state that this was not asked. This section as such it is speculative and not supported by evidence.
Although this dataset does not include data for TSE in pregnancy and in-utero exposure, we attempted to outline similar data in literature through this section
Line 364-368 - IT is not clear what point the authors are trying to make with that paragraph. Could you elaborate more?
Thank you, elaborated in the text
Line 369-375 - It is good that you are considering clinical implications, but it is not clear how you would expect someone to integrate, could you provide examples of what that would look like? Any successful examples/citations of what you are suggesting? This section comes across as an afterthought and for someone that is working in behaviour change and developing interventions this section is not adding something that is not already known.
Added separate paragraph for harm reduction
Limitation Section. your limitation section does not take into account some key methodological limitations of your research. Please review.
- Added methodological limitations
Reviewer 4 Report
Comments and Suggestions for Authors
This is a typical cross-sectional study.
1. The study aim is correct.
2. The language used in this study is correct.
3. Research topic is novel and important.
4. The study design is correct, as the authors used data from the NSCH study
5. Research methods and statistical analysis are correct.
Minor changes are needed:
1. Please provide more data on why the NSCH study was chosen as a data source.
2. Table 3 should be revised to provide clarity of the text
3. Discussion is a little bit extensive and should focus more on own findings and key observations.
4. 3-4 sentences on the practical implications of this study (e.g., placed before the limitations section) will increase the practical application of the findings presented by the Authors
Author Response
Thank you for reviewing our work and providing your feedback. Please find below, responses in italics and changes made to the revised draft submitted with MDPI:
This is a typical cross-sectional study.
- The study aim is correct.
- The language used in this study is correct.
- Research topic is novel and important.
- The study design is correct, as the authors used data from the NSCH study
- Research methods and statistical analysis are correct.
Minor changes are needed:
- Please provide more data on why the NSCH study was chosen as a data source
Thank you for your question. The NSCH (National Survey of Children's Health) study was chosen as our data source due to its comprehensive and nationally representative dataset. The NSCH provides extensive data on various health, behavioral, and socio-demographic factors across different age groups of children in the United States, making it an ideal source for analyzing both prevalence and comorbidity of psychiatric conditions. Furthermore, the survey's robust methodology and large sample size ensure reliable and generalizable results, which are crucial for the validity of our findings
- Table 3 should be revised to provide clarity of the text
We tried to make it more clear.
- Discussion is a little bit extensive and should focus more on own findings and key observations.
Thank you, amde suggested changed
- 3-4 sentences on the practical implications of this study (e.g., placed before the limitations section) will increase the practical application of the findings presented by the Authors
Added harm reduction strategies